# Ocular Melanoma: A Comprehensive Review with a Focus on Molecular Biology

**DOI:** 10.3390/ijms26199799

**Published:** 2025-10-08

**Authors:** Lucia Iavarone, Renato Franco, Federica Zito Marino, Giuseppe D’Abbronzo, Giuseppe Argenziano, Camila Scharf, Grazia Nucci, Andrea Ronchi, Gerardo Cazzato

**Affiliations:** 1PhD Course in Public Health, Department of Experimental Medicine, University of Campania “Luigi Vanvitelli”, 80138 Naples, Italy; lucia.iavarone@unicampania.it; 2Pathology Unit, Department of Mental and Physical Health and Preventive Medicine, University of Campania “Luigi Vanvitelli”, Via Luciano Armanni 5, 80138 Naples, Italy; renato.franco@unicampania.it (R.F.); federica.zitomarino@unicampania.it (F.Z.M.); dabbronzogiuseppe@gmail.com (G.D.); 3Dermatology Unit, Department of Mental and Physical Health and Preventive Medicine, University of Campania “Luigi Vanvitelli”, 80138 Naples, Italy; giuseppe.argenziano@unicampania.it (G.A.); camila.araujoscharfpinto@unicampania.it (C.S.); 4Section of Molecular Pathology, Department of Precision and Regenerative Medicine and Ionian Area (DiMePRe-J), University of Bari “Aldo Moro”, 70121 Bari, Italy; grazia.nucci@asl.brindisi.it (G.N.); gerardo.cazzato@uniba.it (G.C.)

**Keywords:** ocular melanoma, eye melanoma, uveal melanoma, conjunctival melanoma, eyelid melanoma

## Abstract

Ocular melanoma is a rare but clinically significant malignancy, primarily comprising uveal and conjunctival subtypes. Although sharing some histopathological features with cutaneous melanoma, these tumours are characterized by distinct molecular and biological profiles with direct implications for prognosis and treatment. Uveal melanoma is predominantly driven by mutations in *GNAQ* and *GNA11*, along with alterations in *BAP1*, *SF3B1*, and *EIF1AX*, which are key prognostic determinants. Conversely, conjunctival and eyelid melanoma exhibits greater molecular similarity to cutaneous melanoma, commonly involving *BRAF*, *NRAS*, *NF1*, and *TERT* promoter mutations. Despite progress in the molecular characterization of these entities, metastatic disease continues to confer a poor prognosis, particularly in uveal melanoma. Ongoing research into the molecular basis of ocular melanoma is essential to advance targeted therapies and improve clinical outcomes. The aim of this review is to provide a comprehensive overview of ocular melanoma, with a particular focus on the molecular biology underlying its clinical behaviour and emerging therapeutic opportunities.

## 1. Introduction

Ocular melanoma represents a heterogeneous group of malignancies arising from melanocytic cells within the eye, encompassing uveal melanoma, conjunctival melanoma, and eyelid melanoma. Although collectively rare compared to cutaneous melanoma, ocular melanomas pose significant clinical challenges due to their unique anatomical locations, diverse biological behaviour, and limited therapeutic options for metastatic disease. Among these, uveal melanoma constitutes the majority of cases, arising primarily from melanocytes of the choroid, ciliary body, and iris. Conjunctival melanoma, in contrast, originates from melanocytes of the bulbar or palpebral conjunctiva and is epidemiologically and molecularly closer to cutaneous melanoma. Eyelid melanoma, although infrequent, shares histopathological characteristics with both cutaneous and conjunctival melanomas, reflecting the complex interplay between anatomical site and oncogenic pathways.

The molecular landscape of ocular melanoma is distinct and increasingly recognized as a critical determinant of prognosis and therapeutic responsiveness [1]. Uveal melanoma is characterized by a near-universal absence of *BRAF* mutations, which are prevalent in cutaneous melanoma, and instead exhibits recurrent activating mutations in *GNAQ* and *GNA11*, leading to constitutive MAPK and YAP pathway activation [2,3,4,5,6]. Additional genetic alterations in *BAP1*, *SF3B1*, and *EIF1AX* are strongly associated with metastatic risk and clinical outcomes, highlighting the value of molecular profiling in risk stratification. By contrast, conjunctival melanoma frequently harbours mutations in *BRAF, NRAS*, and *TERT* promoter regions, reflecting its partial overlap with cutaneous melanoma oncogenesis, while also demonstrating UV-induced mutational signatures, particularly in exposed areas of the ocular surface [7,8]. Eyelid melanoma, although less extensively characterized, appears to share mutational patterns with both cutaneous and conjunctival counterparts, with *BRAF* and *NRAS* mutations reported in a subset of cases [1].

Beyond single-gene alterations, emerging evidence underscores the importance of epigenetic modifications, copy number variations, and transcriptomic profiles in defining tumour behaviour and therapeutic vulnerabilities. Given these divergent molecular features, ocular melanoma exemplifies a model in which site-specific biology profoundly influences both clinical management and translational research strategies. The identification of molecular subtypes has informed prognostic classification, guided the development of novel targeted therapies, and facilitated patient selection for clinical trials, particularly in metastatic settings where conventional treatments are largely ineffective. Moreover, understanding the interplay between mutational burden, immune microenvironment, and therapeutic response remains a critical frontier, with implications for immunotherapy and precision oncology approaches.

This review aims to synthesize current knowledge on the molecular underpinnings of ocular melanoma across uveal, conjunctival, and eyelid subtypes, highlighting key genetic and epigenetic alterations, their prognostic significance, and emerging therapeutic opportunities. By integrating findings from recent genomic, transcriptomic, and epigenomic studies, we provide a comprehensive framework for understanding the molecular heterogeneity of ocular melanoma, thereby supporting the development of rational, mechanism-based clinical strategies.

## 2. Uveal Melanoma

### 2.1. Epidemiology

Uveal melanoma is the most common primary intraocular malignancy in adults, although it remains a relatively rare cancer overall. Its incidence is estimated at approximately 4 to 6 cases per million individuals annually in the United States and Europe [9]. The disease occurs more frequently in Northern European populations, while it is markedly less common among individuals of Asian or African descent [10]. Uveal melanoma typically presents between the ages of 50 and 70 years, with a slight male predominance [11]. It is especially prevalent among Caucasians, particularly those with light-coloured eyes (blue or green) and fair skin [12]. These phenotypic traits, in addition to predisposing conditions such as ocular or oculo-dermal melanocytosis and dysplastic nevus syndrome, are recognized risk factors. In contrast to cutaneous melanoma, however, a definitive association with ultraviolet (UV) radiation exposure has not been established.

From an anatomical perspective, uveal melanoma arises from melanocytes within the uveal tract, comprising the choroid, ciliary body, and iris. The choroid represents the predominant site of origin, accounting for 85–90% of cases, followed by the ciliary body (5–8%) and the iris (3–5%) [13]. Iris melanomas are generally detected at earlier stages and often display a more indolent clinical behaviour.

Despite significant progress in local therapeutic approaches, approximately half of patients with uveal melanoma eventually develop metastatic disease, most commonly to the liver [14]. The prognosis in metastatic cases remains poor, with a median survival of only 6 to 12 months [15]. Several clinicopathological and molecular factors influence patient outcomes, including tumour size and location, histopathological subtype (epithelioid morphology conferring a worse prognosis compared with spindle cell tumours), and specific cytogenetic or molecular alterations, such as monosomy 3 and *BAP1* mutations [16,17,18].

Uveal melanoma thus remains a formidable clinical challenge, characterized by insidious progression, the potential for delayed metastatic spread, and the absence of effective systemic treatment options for advanced disease.

### 2.2. Molecular Biology

Uveal melanoma displays unique molecular and genetic hallmarks that clearly distinguish it from cutaneous melanoma. These features not only provide insight into tumour biology and prognosis but also represent potential avenues for targeted therapeutic intervention.

One of the most defining molecular events in uveal melanoma is the presence of activating mutations in the G-protein alpha subunit genes, *GNAQ* and *GNA11* [1,2,3]. Such mutations, which occur in more than 90% of cases, are regarded as early, initiating drivers of tumorigenesis [19]. They result in constitutive activation of the MAPK and YAP signalling cascades, thereby promoting uncontrolled cell proliferation and survival [20].

At the mechanistic level, mutant *GNAQ* and *GNA11* proteins stimulate phospholipase C (PLC), leading to the hydrolysis of phosphatidylinositol diphosphate into inositol triphosphate (IP3) and diacylglycerol (DAG). These second messengers activate calcium-dependent pathways and protein kinase C (PKC). PKC, in turn, initiates the MAPK signalling cascade through sequential phosphorylation of Raf, MEK1/2, and ERK, ultimately driving overexpression of the cell cycle regulator cyclin D1 (CCND1). This downstream signalling culminates in the modulation of transcription factors that govern both proliferation and apoptosis.

Interestingly, studies have shown that MAPK pathway activation is not uniform, but rather heterogeneous, across primary uveal melanomas harbouring *GNAQ* or *GNA11* mutations [19]. Importantly, mutations in these two genes are mutually exclusive and typically occur in a heterozygous state [4,5,6].

The *GNAQ* and *GNA11* genes are located on chromosomes 9q21.2 and 19p13.3, respectively. They are paralogous genes, derived from a common ancestral gene, and share approximately 90% sequence homology [21]. Each gene contains seven coding exons and encodes a protein of nearly identical size (≈42 kDa) and length (359 amino acids). These proteins correspond to the αq and α11 subunits of heterotrimeric G proteins, which are key mediators of intracellular signalling [21]. The schematic representation of GNAQ/11 pathways is showed in Figure 1.

Mutations in *GNAQ* and *GNA11* represent the most frequent genetic alterations in uveal melanoma, occurring in 80–90% of cases. Reported mutation rates for *GNAQ* range from 24.2% to 53.3%, while those for *GNA11* vary between 24.2% and 60% [3,22]. Although early studies suggested lower frequencies in non-Caucasian populations, more recent data indicate that mutation prevalence is broadly similar across ethnic groups, underscoring the need to further investigate the interplay of demographic and genetic factors [19].

Most pathogenic variants in *GNAQ* and *GNA11* are activating missense mutations clustering within exons 4 and 5, particularly at codons 183 (R183) and 209 (Q209) [23,24]. Q209 mutations are more frequent and typically result from single-nucleotide substitutions at the CAG codon, leading to replacement of glutamine (Q) with leucine (L) or proline (P). The most common base changes involve A > T and A > C transitions. Less common exon 5 variants include p.Q209M, p.Q209H, p.Q209I, p.F228L, and p.M203V in *GNAQ*, as well as p.Q209Y, p.E234K, and p.E221D in *GNA11* [25,26,27].

Exon 4 mutations are less frequent. In *GNA11*, substitutions at codon 183 (CGC) typically arise from C > T transitions, producing p.R183C or p.R183H. Similarly, *GNAQ* mutations at codon 183 (CGA) usually result from G > A transitions [3]. Other rare *GNAQ* exon 4 variants include p.P170S, p.I189T, p.Q176R, and p.P193L, which together account for approximately 8.9% of cases in some series [25].

Functionally, mutations at codons 183 and 209 alter GTPase activity in distinct ways. Q209 variants abolish GTPase function entirely, leading to constitutive activation of the Gα subunit and persistent downstream signalling, whereas R183 mutations result in partial GTPase loss, generating weaker but still pathogenic signalling [19]. Specifically, R183 and Q209 substitutions affect the switch I and switch II domains of the Gq/11 proteins, respectively, locking the α subunit in an active, GTP-bound state and thereby driving continuous, receptor-independent signalling [19].

In addition to *GNAQ* and *GNA11*, recurrent mutations have been identified in *PLCB4*, a downstream effector of *GNAQ/GNA11*. The hotspot mutation occurs within the highly conserved catalytic Y-domain, which is critical for intracellular transduction of extracellular cues, particularly in the retina. This gain-of-function alteration enhances *PLCB4* activity and converges on the same signalling cascade activated by *GNAQ/GNA11* mutations [28].

As uveal melanoma progresses, secondary mutations and chromosomal alterations become critical determinants of prognosis and metastatic potential. Among these, monosomy 3 is one of the most powerful prognostic markers, strongly associated with an increased risk of metastasis [29,30]. Approximately half of uveal melanomas display monosomy 3, a feature frequently linked to mutations or loss of expression of the *BAP1* (BRCA1-associated protein 1) gene [31]. *BAP1* alterations—detected in up to 80% of monosomy 3 tumours—correlate with aggressive biological behaviour and poor clinical outcome. Located on chromosome 3p21.1, *BAP1* encodes a deubiquitinating enzyme that regulates protein stability and transcriptional programmes involved in melanocyte differentiation and function. Inactivating somatic mutations in *BAP1* promote dedifferentiation and drive a metastatic phenotype; indeed, up to 84% of metastasizing uveal melanomas harbour *BAP1* loss-of-function variants. Beyond uveal melanoma, *BAP1* inactivation has been implicated in several malignancies, including breast and lung cancer. Germline mutations define the *BAP1* tumour predisposition syndrome, conferring increased susceptibility to multiple cancers, including uveal melanoma [32].

Conversely, tumours retaining both copies of chromosome 3 (disomy 3) and exhibiting chromosome 6p gain generally have a more favourable prognosis [33]. This genetic profile is frequently associated with *EIF1AX* and *SF3B1* mutations [34].

*EIF1AX* mutations are typically restricted to disomic, low-risk tumours and occur in the highly conserved amino-terminal region of the gene. These missense variants, found in approximately 18% of cases, are mutually exclusive with *BAP1* and *SF3B1* mutations [35,36]. Functionally, *EIF1AX* encodes a factor essential for translation initiation, facilitating the recruitment of Met-tRNAi to the ribosome. Although the precise consequences of these mutations remain uncertain, they may alter start codon recognition. Clinically, *EIF1AX* mutations are consistently associated with indolent disease and a low risk of metastasis, despite contributing to tumorigenesis.

*SF3B1* mutations define an intermediate-risk subset of uveal melanoma. They are usually observed in disomic tumours lacking *BAP1* mutations and are most commonly located at codons R625 and K666 [37]. These mutations disrupt normal RNA splicing by favouring the use of cryptic splice sites, thereby altering the transcriptional landscape. Clinically, *SF3B1*-mutant tumours are associated with late-onset metastasis, distinguishing them from the early-metastatic *BAP1*-mutant tumours. A defining feature of this subclass is overexpression of PRAME (Preferentially Expressed Antigen in Melanoma), a transcriptional repressor of retinoic acid receptor signalling. PRAME expression, regulated by promoter methylation, has emerged as an additional biomarker of metastatic risk in both *SF3B1*- and *BAP1*-mutant tumours [37].

Unlike cutaneous melanoma, uveal melanoma rarely harbours mutations in *BRAF* or *NRAS*, explaining the lack of efficacy of conventional BRAF/MEK inhibitors [38]. Nevertheless, elevated levels of phosphorylated MEK and ERK are consistently observed in primary and metastatic tumours, even in the absence of *BRAF* mutations [19]. This underscores the unique molecular profile of uveal melanoma, which presents major challenges for systemic therapy. Current investigational approaches include inhibitors targeting YAP and G-protein signalling, as well as immunotherapeutic strategies tailored to the tumour’s genetic background [39,40].

In summary, uveal melanoma is initiated by early *GNAQ/GNA11* mutations and subsequently shaped by additional genetic and chromosomal events, most notably alterations in *BAP1*, *SF3B1*, and *EIF1AX*, along with monosomy 3 and 6p gain. These molecular features are central to prognostication and increasingly represent the foundation for novel therapeutic strategies.

### 2.3. Histological Features

Uveal melanoma exhibits a wide spectrum of cytomorphologic and architectural features, which play a pivotal role in both diagnosis and prognostication. Based on cytomorphology, tumour cells are classically divided into three categories: spindle cell, epithelioid cell, and mixed types [32].

Spindle cells are elongated and fusiform, with slender nuclei containing finely dispersed chromatin and inconspicuous nucleoli, particularly in spindle A cells [41]. Spindle A cells typically show low mitotic activity and minimal pleomorphism, correlating with a more favourable prognosis. In contrast, spindle B cells display greater nuclear irregularity and increased mitotic activity.

Epithelioid cells are large and polygonal, with abundant eosinophilic cytoplasm, prominent nucleoli, and pronounced nuclear pleomorphism and hyperchromasia. Their presence is strongly associated with aggressive biological behaviour and poor clinical outcome. Mixed tumours, containing variable proportions of spindle and epithelioid cells, demonstrate an intermediate prognosis [42].

Additional histopathological features that support a diagnosis of melanoma include a high mitotic index (often > 1 mitosis per 40 high-power fields), marked nuclear pleomorphism, an infiltrative growth pattern with extension beyond the uveal tract, and the presence of closed vascular loops and networks highlighted by periodic acid–Schiff (PAS) staining. These vasculogenic patterns represent distinctive microcirculatory arrangements that are strongly linked to metastatic potential [43].

By contrast, uveal nevi are benign melanocytic proliferations typically composed of uniform spindle cells with minimal cytologic atypia and absent or very low mitotic activity. Nevi generally exhibit well-circumscribed, non-infiltrative growth confined to the uveal stroma. Nuclear morphology is bland, with small, regular nuclei and inconspicuous nucleoli, while epithelioid cells are uncommon or absent.

Differential diagnosis is summarized in Table 1.

Immunohistochemistry confirms the melanocytic nature of the lesion but lacks reliability in differentiating benign from malignant entities. Both uveal nevi and melanomas typically express Melan-A, HMB-45, and S100. However, Ki-67 labelling indices are significantly higher in melanomas, reflecting increased proliferative activity and showing prognostic relevance [44].

The diagnostic and prognostic utility of PRAME expression in distinguishing uveal melanoma from uveal melanocytic nevi remains under investigation, and current data are still inconclusive [45,46]. In morphologically ambiguous cases, molecular genetic testing—such as detection of monosomy 3 or BAP1 loss—can provide valuable adjunctive information, complement histopathologic assessment, and contribute to risk stratification.

### 2.4. Prognosis

Approximately half of patients with uveal melanoma eventually develop metastatic disease, most commonly within 15 years of the initial diagnosis [47]. Once distant metastasis occurs, the prognosis is dismal, with a median survival of less than one year despite the use of systemic therapies. This poor outcome reflects the intrinsic resistance of uveal melanoma to conventional chemotherapy and the current lack of highly effective systemic treatment options.

Prognostic determinants include tumour size, anatomic location (with ciliary body involvement conferring an adverse prognosis), cytomorphologic subtype (epithelioid morphology being associated with worse outcomes than spindle morphology), and underlying genetic alterations [16]. Among these, monosomy 3 and *BAP1* mutations are the most powerful predictors of metastatic progression and reduced overall survival [48,49]. Conversely, patients whose tumours retain both copies of chromosome 3 and lack *BAP1* mutations generally experience a more favourable clinical course.

In recent years, molecular classification through gene expression profiling has emerged as a crucial tool for stratifying metastatic risk and guiding patient management [50,51,52].

## 3. Conjunctival Melanoma

### 3.1. Epidemiology

Conjunctival melanoma is an uncommon yet potentially aggressive malignancy originating from melanocytes within the conjunctival epithelium. Its annual incidence is estimated at 0.2–0.8 cases per million individuals, positioning it among the rarest ocular tumours [53]. The disease predominantly affects individuals of Caucasian descent and is most frequently diagnosed in middle-aged to older adults, although cases in younger patients have also been reported [53]. While conjunctival melanoma can arise de novo, the majority of cases develop from pre-existing lesions [54]. Approximately 75% of tumours originate from primary acquired melanosis (PAM) with atypia, around 20% emerge from pre-existing conjunctival nevi, and only 5–10% appear without any identifiable precursor lesion [55].

Clinically, conjunctival melanomas most commonly involve the bulbar conjunctiva, particularly within the interpalpebral zone [53]. Nevertheless, they may extend to the cornea, fornix, caruncle, or eyelid, and in advanced cases, invade the orbit. Despite often presenting at an early stage, these tumours carry a high risk of local recurrence and distant metastasis [56,57,58]. Local recurrence rates have been reported to reach up to 50%, while metastatic disease develops in approximately 20–40% of patients, predominantly affecting regional lymph nodes, lungs, liver, and brain. The 10-year survival rate is variable, estimated at 70–80% in non-metastatic cases, but declines sharply once systemic dissemination occurs. Geographically, the incidence of conjunctival melanoma is higher in Northern Europe, North America, and Australia, whereas it is exceedingly rare among individuals of African or Asian descent [56,57,58].

### 3.2. Molecular Biology

Unlike uveal melanoma, which is predominantly driven by *GNAQ* and *GNA11* mutations, conjunctival melanoma exhibits molecular features more akin to cutaneous melanoma [7]. A hallmark of its molecular landscape is the presence of activating mutations in the MAPK signalling pathway, particularly in the *BRAF* and *NRAS* genes. *BRAF* mutations, most frequently the V600E variant, are detected in approximately 30–50% of cases, followed by V600K (≈20%) and less common variants such as V600D and V600R [59,60]. These alterations result in constitutive MAPK/ERK pathway activation, thereby promoting tumour cell proliferation and survival. The schematic representation of BRAF pathways is shown in Figure 2.

Notably, MEK, ERK, and AKT signalling in conjunctival melanoma tissue is not solely dependent on *BRAF* mutational status, indicating the contribution of additional mechanisms to MAPK and PI3K-AKT pathway activation.

Although *BRAF* mutations are present in 19–56% of conjunctival nevi—potentially at higher frequencies than in conjunctival melanoma—the degree of MAPK pathway activation is more pronounced in malignant tumours, supporting the involvement of alternative drivers such as *NF1* mutations. Interestingly, the high prevalence of *BRAF* mutations in conjunctival nevi parallels that observed in cutaneous nevi, where one study reported a rate of 82% [61]. *NRAS* mutations occur in approximately 10–20% of cases, similarly driving MAPK pathway activation [62]. Oncogenic *RAS* mutations in human melanomas typically involve codons 12, 13, or 61. Among *NRAS*-mutant tumours, the *NRAS*^Q61 mutation is present in 90% of cases, promoting constitutive GTP-bound RAS activity, whereas codon 12 or 13 mutations primarily impair GTP hydrolysis. Specifically, the *NRAS*^Q61R substitution exhibits greater intrinsic tumorigenic potential in melanocytes than *NRAS*^G12D. Unlike normal melanocytes, *NRAS*-mutant melanomas activate the MAPK pathway predominantly via the *NRAS* effector *CRAF* rather than *BRAF*. Although prior studies have suggested concurrent involvement of MAPK and PI3K signalling, MAPK-mediated signalling predominates in *NRAS*-mutant tumours [63].

Additionally, the Neurofibromin 1 (*NF1*) gene (17q11.2) encodes a GTPase-activating protein that inhibits *RAS* by promoting GTP hydrolysis. *NF1* mutations, leading to loss of neurofibromin function, are observed in up to 30% of conjunctival melanomas and may co-occur with *BRAF* or *NRAS* mutations [64].

In addition to MAPK pathway alterations, telomerase reverse transcriptase (*TERT*) promoter mutations are frequently observed in conjunctival melanomas and are associated with increased telomerase activity and cellular immortality [65]. These mutations are considered markers of tumour aggressiveness and poor prognosis [8]. TERT, the catalytic subunit of the telomerase complex, is activated downstream of the AKT pathway and mediates the elongation of telomeres by adding repetitive TTAGGG sequences to chromosomal ends. This protects chromosomes from degradation and enables sustained cellular proliferation, thereby contributing to the acquisition of cellular immortality. *TERT* promoter mutations, which increase *TERT* expression, have been identified in approximately 32–41% of conjunctival melanomas and in 8% of primary acquired melanosis (PAM) cases. In conjunctival melanoma, these mutations predominantly involve C > T or CC > TT transitions, characteristic of ultraviolet (UV) radiation-induced DNA damage, suggesting a potential etiological role of UV exposure in tumour development [64].

*KIT* mutations are less common, reported in 2–7% of cases [64]. The *KIT* gene, located on chromosome 4q12, encodes a receptor tyrosine kinase (RTK) that activates multiple downstream signalling cascades, including MAPK and PI3K/AKT/mTOR, upon ligand binding. Oncogenic *KIT* mutations in conjunctival melanoma result in constitutive receptor activation and hyperactivation of these pathways. Although *KIT* mutations are infrequent, overexpression of *KIT* is more common, likely due to copy number alterations. Notably, tumours harbouring *KIT* mutations or amplification generally lack *BRAF* or *NRAS* mutations, indicating mutual exclusivity between these genetic alterations [66].

Other relevant molecular events include mutations or deletions in tumour suppressor genes such as *TP53* and *CDKN2A*, which promote uncontrolled cell cycle progression and genomic instability [61,66,67]. *CDKN2A*, located at 9p21.3, encodes the p16^INK4a^ protein, a key inhibitor of the cyclin-dependent kinase 4/6 (CDK4/6)-cyclin D complex. Inactivating mutations impair p16 function, allowing unchecked CDK4/6-cyclin D activity during G1-to-S phase progression and increasing mitotic activity. *CDKN2A* alterations have been documented in both cutaneous and conjunctival melanomas. Germline *CDKN2A* mutations are associated with familial melanoma syndromes. Compared to benign melanocytic lesions and PAM with atypia, conjunctival melanomas typically show reduced nuclear p16 expression [64].

Chromosomal aberrations are also common, with recurrent gains in 1q, 3q, and 6p, and losses in 10q and 11q [68]. Recent evidence links 10q loss to *BRAF* mutations, increased tumour thickness, lymphatic invasion, and metastasis. This region encodes several tumour suppressors, including SUFU, NEURL1, PDCD4, and C10orf90 [61].

These molecular insights have significant therapeutic implications. *BRAF* mutations provide a rationale for targeted therapy with BRAF inhibitors (e.g., vemurafenib or dabrafenib), often combined with MEK inhibitors. Immune checkpoint inhibitors, such as anti-PD-1 or anti-CTLA-4 agents, are also under investigation, particularly in metastatic or unresectable cases, although clinical experience is largely extrapolated from cutaneous melanoma due to the rarity of conjunctival melanoma.

### 3.3. Histological Features

Histologically, conjunctival melanoma is characterized by a proliferation of atypical melanocytes, which may display epithelioid, spindle-shaped, or mixed morphologies. These malignant cells typically exhibit nuclear pleomorphism, prominent nucleoli, increased mitotic activity, and variable cytoplasmic melanin pigmentation [69]. The tumour often demonstrates an infiltrative growth pattern within the conjunctival epithelium and underlying substantia propria, and may show pagetoid spread, in which atypical melanocytes migrate upward through the epithelial layers. Lymphovascular invasion can also occur and is associated with an elevated risk of metastasis [69].

A critical aspect of diagnosis is distinguishing conjunctival melanoma from benign melanocytic lesions, particularly conjunctival nevi, which are far more common and usually benign. Nevi typically present during childhood or adolescence, whereas melanoma most often arises in middle-aged or older adults. Histologically, nevi consist of well-circumscribed nests or lobules of uniform, small, round-to-oval melanocytes within the subepithelial stroma. These cells generally display minimal atypia, low mitotic activity, and no pagetoid spread. Cystic epithelial inclusions, frequently observed in nevi, are a key diagnostic feature absent in melanomas. In contrast, conjunctival melanomas lack cystic inclusions, exhibit cytologic atypia, and demonstrate an invasive growth pattern [70]. Additional histologic features supportive of malignancy include mitotic figures, ulceration, and extension beyond the basal epithelial layer.

Immunohistochemistry can aid in the differential diagnosis. Both nevi and melanomas are typically positive for melanocytic markers such as S-100, HMB-45, and Melan-A; however, a high Ki-67 proliferation index favours a diagnosis of melanoma [71]. Emerging data suggest that PRAME may serve as a useful adjunct marker to distinguish conjunctival melanoma from nevi [72].

Ultimately, accurate diagnosis relies on an integrated assessment of histologic features, clinical history—including rapid growth or morphological changes—and, when indicated, adjunct molecular or immunohistochemical testing. Correctly differentiating conjunctival melanoma from benign melanocytic lesions is essential, as it directly informs prognosis and clinical management.

### 3.4. Prognosis

The prognosis of conjunctival melanoma varies considerably and is influenced by multiple clinical and pathological factors. Overall, long-term outcomes are guarded, particularly in cases exhibiting high-risk features. Tumour location is a critical determinant of prognosis: lesions involving the fornix or palpebral conjunctiva generally have a poorer outcome compared with those confined to the bulbar conjunctiva. Tumour thickness and size are also important prognostic indicators, with larger and thicker lesions associated with increased risks of recurrence and metastasis. Histopathological parameters, including lymph vascular invasion and mitotic activity, further modulate the risk of disease progression [73,74,75,76].

Local recurrence is relatively common, necessitating prolonged surveillance. Even after complete surgical excision, recurrence rates range from 20% to over 50%, particularly when margins are positive or disease is multifocal [77,78]. Regional lymph nodes are frequent initial sites of metastasis, although distant dissemination—most commonly to the lungs, liver, and brain—can also occur. Approximately 15–25% of patients eventually develop metastatic disease, typically within five years of diagnosis, though late metastases have been reported [74].

Survival outcomes are heterogeneous. Five-year disease-specific survival rates have been reported between 70% and 85%, with declines observed in patients harbouring high-risk features [73,74,75,76]. Close clinical follow-up, including imaging and sentinel lymph node biopsy in selected cases, is recommended to facilitate early detection of recurrence or metastasis. Treatment strategies continue to evolve, with some centres investigating adjuvant approaches such as topical chemotherapy, cryotherapy, or targeted systemic therapies in advanced cases, although robust evidence is limited due to the rarity of the disease.

In summary, conjunctival melanoma carries a substantial risk of local recurrence and distant metastasis. Prognosis is primarily determined by tumour location, size, histological characteristics, and completeness of initial treatment. Long-term surveillance is essential, given the potential for late recurrence and systemic spread.

## 4. Eyelid Melanoma

### 4.1. Epidemiology

Eyelid melanoma represents less than 1% of all cutaneous melanomas and approximately 1% of eyelid malignancies [79,80]. Although rare, it presents a significant clinical challenge due to its potential for local invasion, distant metastasis, and the anatomical and functional complexity of the eyelid region. Epidemiological studies indicate that eyelid melanoma predominantly affects older adults, typically between the sixth and eighth decades of life, with some reports suggesting a slight female predominance, although this may vary according to geographic region and population [79,80]. The annual incidence is estimated at 0.05–0.1 cases per 100,000 individuals. The condition is more commonly diagnosed in individuals with lighter skin phototypes, particularly Fitzpatrick types I and II, consistent with the general epidemiology of cutaneous melanoma [79,80,81]. The lower eyelid is more frequently involved than the upper eyelid, likely reflecting greater cumulative exposure to ultraviolet radiation [82,83].

### 4.2. Molecular Biology

The molecular landscape of eyelid melanoma largely parallels that of cutaneous melanomas arising in other sun-exposed regions, reflecting their shared melanocytic origin and similar environmental risk factors, particularly ultraviolet (UV) radiation [83]. However, due to the rarity of eyelid melanoma, dedicated molecular studies are limited, and most insights are extrapolated from broader investigations of cutaneous melanoma, especially those involving facial or head and neck sites.

One of the most prominent molecular features of eyelid melanoma, particularly in lentigo maligna melanoma—the predominant subtype in this region—is the presence of mutations in the *NRAS* and *BRAF* genes, although overall mutation rates are generally lower than in melanomas of the trunk or limbs [83]. *BRAF* mutations, most notably the V600E variant, are less frequent in chronically sun-damaged facial skin, including the eyelids, than in melanomas of intermittently sun-exposed skin. Approximately 50% of cutaneous melanomas harbour *BRAF* mutations, predominantly at codon 600 within the kinase activation domain, where valine is replaced by glutamic acid (*BRAF* V600E), resulting in a T-to-A nucleotide substitution. This single amino acid change constitutively activates the kinase, driving phosphorylation of MEK, a downstream effector in the MAPK pathway, which in turn phosphorylates extracellular signal-regulated kinase (ERK). ERK modulates numerous cytoplasmic substrates and transcription factors, including Ets-1 and Elk. Hyperactivation of the MAPK pathway contributes to the oncogenic phenotype by promoting uncontrolled proliferation (via sustained cyclin D1 expression), inhibiting apoptosis (through suppression of pro-apoptotic proteins such as BIM), and facilitating tumour invasion (via maintenance of integrin expression and cytoskeletal remodelling). Importantly, the T-to-A transition is not considered UV-induced. Other clinically relevant *BRAF* mutations include V600K (≈20% of *BRAF*-mutant cases) and V600R (5–7%), with evidence suggesting that V600K is associated with older age and chronic sun damage [83].

In contrast, *NRAS* mutations and other alterations linked to chronic sun damage (CSD) are more prevalent in the periorbital region, reflecting cumulative UV exposure [83]. The *RAS* family encodes low-molecular-weight GTP-binding proteins that, upon activation by growth factors or oncogenic mutations, stimulate multiple intracellular signalling pathways, including MAPK and PI3K/AKT, as well as additional effectors such as Ral-GDS and phospholipase C-eta. Mutations have been reported in *NRAS* (20%), *KRAS* (2%), and *HRAS* (1%) in melanoma, with *NRAS* Q61 being the most frequent hotspot [83].

*TERT* promoter mutations, associated with tumour aggressiveness and poor prognosis, may also occur, although their prevalence specifically in eyelid melanoma remains unclear due to limited targeted studies. Tumours from chronically sun-exposed sites often exhibit high mutational burdens with a UV-signature, characterized by C > T transitions at dipyrimidine sites. Emerging technologies such as next-generation sequencing (NGS) are enabling more comprehensive genomic profiling, revealing additional alterations in genes including *KIT*, *TP53*, and *GNAQ*, although their clinical significance in eyelid melanoma is still under investigation.

Notably, point mutations in *TP53* have been detected in 67% of sebaceous eyelid carcinomas. These mutations resemble those observed in internal malignancies, distinguishing them from UV-induced *TP53* mutations typical of cutaneous cancers. *TP53* mutation status appeared independent of tumour stage; however, reduced expression of p21—a transcriptional target of p53—was associated with lymph node metastasis, suggesting that p21 may serve as a potential prognostic marker in sebaceous eyelid carcinomas [84].

In summary, while eyelid melanoma shares many molecular characteristics with melanomas of chronically sun-exposed areas, its unique anatomical context and low incidence underscore the need for dedicated studies to determine whether distinct molecular features could inform prognosis or guide targeted therapies.

### 4.3. Histological Features

Histologically, eyelid melanoma shares many characteristics with cutaneous melanomas at other anatomical sites; however, its diagnosis can be particularly challenging due to the complex anatomy of the eyelid and the wide histologic spectrum of benign melanocytic lesions in this region. A major diagnostic challenge lies in distinguishing eyelid melanoma—especially lentigo maligna melanoma, the most frequent subtype in this area—from benign melanocytic nevi, including junctional, compound, and intradermal nevi [85].

In its in situ form, lentigo maligna typically exhibits proliferation of atypical melanocytes along the dermo-epidermal junction in a lentiginous pattern. These melanocytes are often large, with hyperchromatic, irregular nuclei, prominent nucleoli, and variable cytoplasm. Extension along adnexal structures, such as hair follicles and sweat ducts, is common. Pagetoid spread may also be observed, although it is more prominent in superficial spreading melanoma than in lentigo maligna. Dermal invasion, when present, is characterized by nests or sheets of atypical melanocytes that may show mitotic activity, necrosis, or a desmoplastic stromal response [80,86].

By contrast, benign eyelid nevi are generally well-circumscribed and symmetrical, composed of small, uniform melanocytes lacking significant cytologic atypia or mitotic activity. They may be confined to the epidermis (junctional nevi), involve both the epidermis and dermis (compound nevi), or be limited to the dermis (intradermal nevi). Intradermal nevi are particularly common on the eyelids and can clinically and histologically mimic early invasive melanoma, especially in the presence of inflammation, fibrosis, or secondary changes such as regression [87].

Architectural disorder and cytologic atypia are crucial distinguishing features of melanoma. Whereas nevi typically demonstrate orderly nesting with progressive maturation of melanocytes as they descend into the dermis, melanoma lacks maturation and often shows disorganized nests or single melanocytes infiltrating the epidermis and dermis. Additionally, melanomas frequently exhibit confluence of junctional nests and irregular distribution, in contrast to the discrete, regularly spaced nests seen in nevi.

Immunohistochemistry can be valuable in challenging cases. Both nevi and melanomas are usually positive for melanocytic markers such as S100, SOX10, and Melan-A, but proliferation markers like Ki-67 may aid differentiation: a high Ki-67 index favours melanoma. Ultimately, diagnosis relies on a combination of clinical history, dermoscopic evaluation, and meticulous histopathological assessment, with attention to features such as asymmetry, pagetoid spread, lack of maturation, cytologic atypia, and mitotic activity. Given the subtle or overlapping features, particularly in small biopsies, evaluation by an expert dermatopathologist is often essential for accurate diagnosis.

### 4.4. Prognosis

The prognosis of eyelid melanoma is influenced by several clinical and histopathological factors, including tumour thickness, ulceration, mitotic rate, and the presence of regional or distant metastases. Overall, eyelid melanomas exhibit a variable prognosis, which may be slightly less favourable than that of cutaneous melanomas arising in less anatomically complex regions, primarily due to diagnostic challenges and the difficulty of achieving wide surgical margins in the periorbital area. Early detection is critical but is often hindered by the morphologic similarity of melanoma to benign lesions, particularly nevi, which are common on the eyelid and can closely mimic melanoma both clinically and histologically.

Breslow thickness, representing the depth of invasion, is one of the most important prognostic factors. Thin melanomas confined to the epidermis or with minimal dermal invasion generally have an excellent prognosis, especially when completely excised with clear margins. However, eyelid melanomas frequently present as subtle pigmented macules or nodules, sometimes mistaken for nevi or other benign lesions such as seborrheic keratoses or pigmented basal cell carcinomas. Such diagnostic delays can allow tumours to progress to more advanced stages, negatively impacting outcomes [79,88,89,90].

In the context of differentiating melanoma from eyelid nevi, the stakes are high: whereas nevi are benign and carry no malignant potential unless transformation occurs, melanoma poses a significant risk of metastasis and death if not promptly diagnosed and treated. Clinically, distinguishing between benign nevi and early melanoma can be challenging, particularly in elderly patients with chronically sun-damaged skin, where lentigo maligna may develop. Histologically, features such as cytologic atypia, lack of maturation, mitotic activity, and pagetoid spread must be carefully evaluated, as overlooking or misinterpreting these characteristics can result in misdiagnosis and undertreatment.

When diagnosed at an early stage and surgically excised with negative margins, eyelid melanomas generally have a favourable long-term prognosis [88]. Surgical margins of at least 0.5 cm are recommended to minimize the risk of recurrence [91]. Reported five-year survival rates for localized disease can exceed 85–90%, depending on the series and population studied [88,89,90]. In contrast, involvement of regional lymph nodes or orbital structures markedly worsens prognosis, reducing survival rates and increasing the likelihood of recurrence [88,89,90].

In summary, the prognosis of eyelid melanoma is highly dependent on early and accurate diagnosis, which is complicated by its frequent clinical and histological resemblance to benign nevi. Careful assessment, and when appropriate, the use of adjunctive diagnostic tools such as dermoscopy, immunohistochemistry, and expert histopathological review are essential for distinguishing benign from malignant melanocytic lesions and ensuring optimal patient outcomes.

## 5. Molecular Distinctions Between Ocular and Cutaneous Melanomas

Although melanoma is a substantially ubiquitous neoplasm [92,93], skin melanoma is certainly the most frequent form. Cutaneous and ocular melanomas share a common origin in melanocytes but diverge significantly at the molecular level, reflecting their distinct embryologic backgrounds, microenvironments, and mutational landscapes. Cutaneous melanoma is primarily driven by mutations in genes associated with the MAPK pathway. The most frequent events include activating mutations in *BRAF* (particularly V600E), *NRAS*, and less frequently *NF1*, which collectively result in sustained proliferation and survival signalling [94]. Ultraviolet (UV) radiation exposure plays a central role, shaping the high mutational burden and characteristic UV-signature mutations of cutaneous melanoma. Uveal melanoma has a rather different molecular profile, while conjunctival melanoma and eyelid melanoma have characteristics that are relatively more similar to skin melanoma. Uveal melanoma is typically driven by mutually exclusive mutations in *GNAQ* and *GNA11*, with downstream activation of the MAPK and YAP pathways [2,3,4,5]. Additional recurrent alterations involve *BAP1* (associated with poor prognosis), *SF3B1*, and *EIF1AX*, which stratify tumours into distinct prognostic classes [31,32]. Unlike cutaneous melanoma, uveal melanoma exhibits a relatively low mutational burden and lacks a clear etiological link to UV radiation. Conjunctival melanoma shows closer molecular resemblance to cutaneous melanoma, often harbouring *BRAF* or *NRAS* mutations, and is likewise influenced by UV exposure [62,63,64]. This similarity has therapeutic implications, as targeted inhibitors developed for cutaneous melanoma may be effective in selected cases of conjunctival melanoma. Eyelid melanoma, conversely, behaves more like cutaneous melanoma, reflecting its localization and genetic drivers, with UV radiation playing a relevant role [83,84].

Taken together, these molecular differences underscore the heterogeneity of melanoma across anatomical sites. While cutaneous melanoma and eyelid melanoma share UV-related, *BRAF/NRAS*-driven pathways, uveal melanoma is characterized by Gαq signalling alterations and distinct chromosomal aberrations, and conjunctival represents a molecular and clinical bridge between the two. These insights have direct implications for diagnosis, prognostic stratification, and the development of targeted therapies.

The comparison between molecular features of cutaneous melanoma and ocular melanoma are summarized in Table 2.

## 6. Conclusions and Perspectives

Ocular melanomas constitute a biologically and clinically diverse group of malignancies whose molecular features are increasingly integral to diagnosis, prognosis, and therapeutic management. Over the past decade, significant progress in genomic and transcriptomic profiling has uncovered distinct oncogenic drivers across anatomical subtypes. Uveal melanoma has emerged as the archetype of a Gαq-driven tumour, with secondary alterations in BAP1, SF3B1, and EIF1AX defining molecular subclasses of prognostic importance. Conversely, conjunctival and eyelid melanomas display molecular patterns akin to cutaneous melanoma, characterized by BRAF, NRAS, and TERT promoter mutations, thereby allowing translational insights from cutaneous models to inform treatment strategies.

Despite these molecular advances, the outlook for metastatic ocular melanoma remains dismal, particularly for the uveal variant, where conventional systemic therapies and immune checkpoint blockade have achieved only modest success. Novel therapeutic avenues—targeting the YAP/TEAD axis, protein kinase C, and G-protein signalling, as well as developments in adoptive cell therapy and bispecific T-cell engagers—are generating renewed optimism for precision-based interventions. The integration of multi-omic approaches with single-cell and spatial transcriptomic analyses promises to deepen understanding of tumour heterogeneity, microenvironmental dynamics, and mechanisms of immune escape.

Going forward, research efforts should emphasize translational studies that connect specific molecular alterations with actionable therapeutic vulnerabilities, while sustained international collaboration remains crucial given the rarity of these malignancies. Expanding the molecular understanding of ocular melanoma will be key to realizing individualized, mechanism-driven treatments that improve outcomes for patients facing these complex cancers.

## Figures and Tables

**Figure 1 ijms-26-09799-f001:**
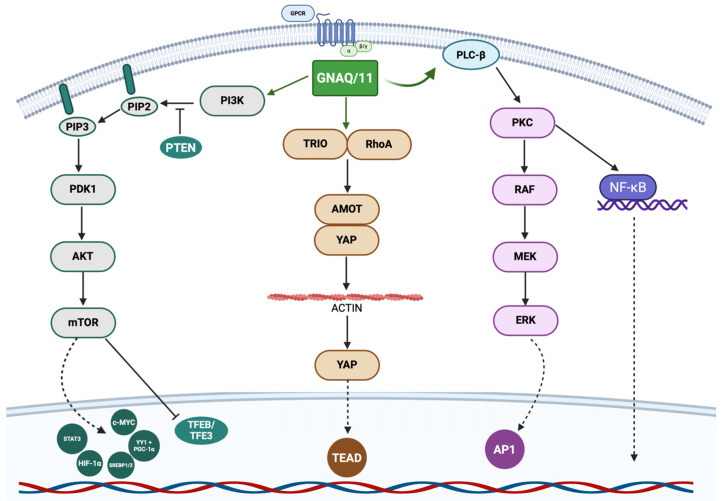
Schematic representation of GNAQ/11 pathways. *GNAQ* and *GNA11* encode Gαq family G-protein alpha subunits, which play a key role in transmitting signals from G-protein–coupled receptors (GPCRs) to intracellular effectors. Binding of GNAQ/11 to the receptor leads to the activation of PLC-β, which recruits and activates protein kinase C (PKC), which in turn triggers the activation of the RAF/MEK/ERK cascade and subsequent induction of AP1 and activation of NF-κB. Activating mutations in *GNAQ/11* maintain chronically active PLC-β, resulting in hyperactivation of PKC–MAPK/ERK and alternative pathways such as YAP/TAZ (Hippo pathway). GNAQ can also directly interact with TRIO, a nucleotide exchange factor (GEF), which, by interacting with RhoA, regulates the cytoskeleton and leads to inhibition of the Hippo cascade, resulting in the association of YAP with the transcription factor TEAD and consequent uncontrolled proliferation, migration, and tumour invasion. In addition to PLC-β and TRIO, GNAQ/11 can activate phosphatidylinositol 3-kinase (PI3K), which phosphorylates phosphatidylinositol-4,5-bisphosphate (PIP_2_) to generate phosphatidylinositol-3,4,5-trisphosphate (PIP_3_). PIP_3_ serves as a recruitment platform for PH-domain-containing proteins, such as PDK1 and AKT. AKT activation, enhanced by mTOR, leads to the phosphorylation of multiple substrates involved in cell survival, growth, and metabolism. Specifically, mTOR activation promotes protein synthesis and proliferation, while inhibition of pro-apoptotic factors promoting tumour survival and growth, in particular it promotes the activation of STAT3, c-MYC, HIF-1α, SREBP 1/2, YY1 + PGC-1α and the inhibition of TFEB/TFE3.

**Figure 2 ijms-26-09799-f002:**
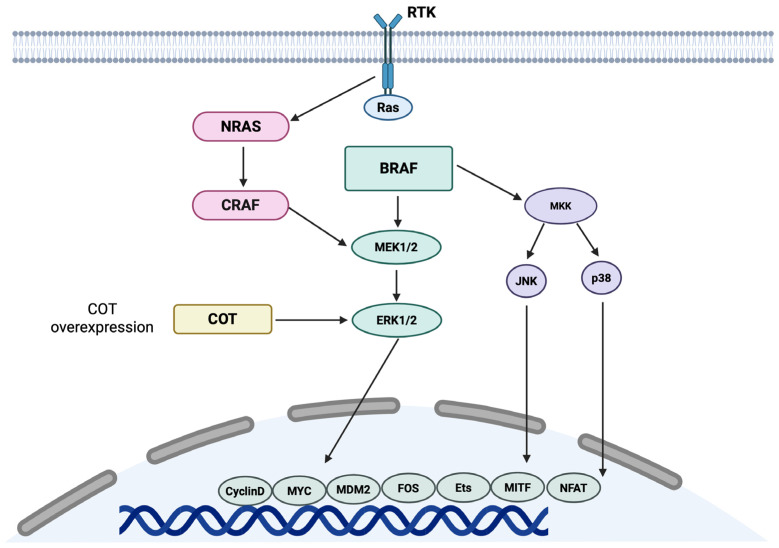
Schematic representation of BRAF pathways. BRAF is a serine-threonine kinase involved in intracellular signal transduction, regulating fundamental processes such as growth, proliferation, differentiation, and cell survival. Like other members of the RAF family (ARAF and CRAF), BRAF is activated by upstream RAS and, in turn, phosphorylates MEK (MAPK/ERK kinase), leading to activation of the ERK (extracellular signal-regulated kinase) pathway. Phosphorylated ERK1/2 subsequently stimulates transcription factors such as CyclinD, MDM2, Ets, MITF, NFAT, Fos and c-Myc, promoting programmes of growth, proliferation, differentiation, migration, angiogenesis, and survival. In parallel, BRAF can activate MKK, which regulates the JNK and p38 pathways, involved in proliferative and stress responses. Under overexpression conditions, COT kinase is able to bypass BRAF activation and directly stimulate ERK1/2, thus contributing to resistance to BRAF-targeted drugs.

**Table 1 ijms-26-09799-t001:** Differential diagnosis of uveal melanocytic proliferations.

Histological Feature	Melanoma	Nevus
Presence of epithelioid cells	Possible	Rare or absent
Mitotic activity	Present	Absent
Nuclear atypia	Marked nuclear enlargement, irregular contours, coarse chromatin, and prominent nucleoli	Mild or absent atypia
Growth pattern	Infiltrative margins with extension into adjacent ocular tissues (e.g., sclera, optic nerve)	Well circumscribed
Closed loops and vascular networks	Present	Absent
Necrosis	May be focally present	Absent

**Table 2 ijms-26-09799-t002:** Molecular features of cutaneous and ocular melanoma.

Feature	Cutaneous Melanoma	Uveal Melanoma	Conjunctival Melanoma	Eyelid Melanoma
Main Driver Mutations	BRAF (V600E), NRAS, NF1	GNAQ, GNA11, BAP1, SF3B1, EIF1AX	BRAF, NRAS, occasional KIT	BRAF, NRAS
Pathways Involved	MAPK, PI3K-AKT	MAPK, YAP/TEAD, splicing alterations	MAPK, PI3K-AKT	MAPK, PI3K-AKT
Mutational Burden	High (UV-signature mutations)	Low	Intermediate	High
Chromosomal Alterations	Variable (chromothripsis, amplifications)	Monosomy 3, chr 8q gain, chr 6p gain	Less defined, some similarities to CM	Variable
Therapeutic Implications	BRAF/MEK inhibitors, immunotherapy	Limited response to immunotherapy; trials with YAP/PKC inhibitors	Potential use of BRAF/MEK inhibitors	BRAF/MEK inhibitors, immunotherapy

## Data Availability

Not applicable: no new data were recorded or analyzed in this study.

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
