# Peer review of "Ocular Melanoma: A Comprehensive Review with a Focus on Molecular Biology"

_ijms, 2025, doi:10.3390/ijms26199799_

Round 1
Reviewer 1 Report
Comments and Suggestions for Authors
This review offers a clear and up-to-date overview of the molecular drivers of ocular melanoma, underlining the relevance of genetic alterations such as GNAQ, GNA11, BAP1, SF3B1, EIF1AX, and TERT promoter mutations for prognosis and patient care. What makes it particularly valuable is the focus on how these biomarkers can refine risk assessment. At the same time, the paper remains quite descriptive, with less emphasis on therapeutic advances and limited comparison between the different subtypes, which slightly narrows its immediate clinical impact.However, the manuscript's integration of genetic insights with prognostic implications makes it a valuable contribution to the field. I recommend acceptance for publication.
Author Response
Thank you for your interest in our work. By incorporating the changes requested by Reviewer 2, we have expanded the comparison between ocular melanoma and cutaneous melanoma, also highlighting the therapeutic implications of molecular aspects, as shown in Table 2.

Reviewer 2 Report
Comments and Suggestions for Authors
The aim of this review is to provide a comprehensive overview of ocular melanoma, with a particular focus on the molecular biology underlying its clinical behavior and emerging therapeutic opportunities.
The review is comprehensive and well written while focusing on a rare type of melanoma.
Comments:
- Gene names should be written in italics.
- Figure 1 should be more detailed, e.g., "TFs" should be exemplified etc.
- The Authors are encouraged to compare the molecular biology of distinct ocular melanomas with skin melanomas. For example, a table comparing molecular and phenotypic aspects can be included.
- Additional figure(s) should be proposed to visualize the major content of this review.
Other aspects of this review are valid and interesting. The current state of the knowledge is fairly demonstrated and discussed. The language is clear and concise. The flow of the manuscript is very good, and the quality of the presentation of the major findings are good.
Author Response
The aim of this review is to provide a comprehensive overview of ocular melanoma, with a particular focus on the molecular biology underlying its clinical behavior and emerging therapeutic opportunities.
The review is comprehensive and well written while focusing on a rare type of melanoma.
Comments:
- Gene names should be written in italics.
AA: we corrected the gene names in italics.
- Figure 1 should be more detailed, e.g., "TFs" should be exemplified etc.
AA: We have redesigned Figure 1 to make it more detailed. The figure legend has also been correspondingly expanded.
- The Authors are encouraged to compare the molecular biology of distinct ocular melanomas with skin melanomas. For example, a table comparing molecular and phenotypic aspects can be included.
AA: We have added a paragraph and a second table dedicated to the comparison of the molecular characteristics of ocular melanoma and cutaneous melanoma.
- Additional figure(s) should be proposed to visualize the major content of this review.
AA: We added a further figure (Figure 2) showing BRAF pathway.
Other aspects of this review are valid and interesting. The current state of the knowledge is fairly demonstrated and discussed. The language is clear and concise. The flow of the manuscript is very good, and the quality of the presentation of the major findings are good.
AA: thank you for your interest in our work and your suggestions.